

**Contrasting trends in element incorporation in hyaline and miliolid foraminifera**
Inge van Dijk[1], Lennart J. de Nooijer[2], Gert-Jan Reichart[1,2]
[1]Department of Ocean Systems, NIOZ-Royal Netherlands Institute for Sea Research, Postbus 59, 1790
AB, Den Burg, the Netherlands, and Utrecht University.
[2]Faculty of Geosciences, Earth Sciences Department, Utrecht University, Budapestlaan 4, 3584 CD,
Utrecht, the Netherlands
**Abstract**
We analyzed trends in element incorporation between hyaline (perforate) and miliolid (imperforate)
foraminifera in order to investigate processes involved in calcification affecting element incorporation
into foraminiferal carbonate. For both groups, we observed similar trends in element incorporation with
$pCO_2$, suggesting there some mechanisms to transports ions to the site of calcification are similar for
both calcification pathways, although the impact might be different across species. A previously
published trans-membrane transport model assumes foraminifera utilize $Ca^{2+}$ channels to transport
calcium to the site of calcification. These channels are somewhat a-specific, leading to (accidental)
transport of other free ions. By modelling the activity of free ions as a function of $pCO_2$, we observed
that speciation of some elements (like Zn and Ba) are heavily influenced by the formation of carbonate
complexes. This leads to an increase in availability of free Zn and Ba with increasing $pCO_2$, which leads
to more transport to the site of calcification and subsequently incorporation in the foraminiferal shell.
We further observed that incorporation of the trace elements studied here is positively correlated
between the hyaline test building species. This could be due to dissimilar activity and/or selectivity of
calcium channels between species, perhaps due to differences in size. For miliolid calcification, part of
the calcium is obtained not only through channels but by also included seawater vesicles, which leads
to similar element to calcium ratios between species and element partitioning which is more in line with
inorganic carbonates.



## 1. Introduction

On the broadest taxonomic scale, calcareous foraminifera, cosmopolitan unicellular protists, produce tests using either one of two fundamentally different mechanisms. These calcification strategies reflect the evolutionary separation of foraminiferal groups dating back to the Cambrian diversification, from where the imperforate miliolids and perforate hyaline foraminifera, developed independently (Pawlowski et al., 2003). The calcification process of the latter group has been studied more extensively than that of the miliolids (De Nooijer et al., 2014). Although many aspects of perforate calcification remain unsolved, there is consensus that chamber formation takes place extracellularly, but within a (semi-) enclosed space, generally termed the site of calcification (SOC). The first layers of calcite precipitate on an organic matrix (the POS or primary organic sheet) that serves as a template for the calcite layer that forms the chamber wall (Hemleben et al., 1977; Erez, 2003). To promote calcification, foraminifera furthermore need to remove Mg ions and/or protons (Zeebe and Sanyal, 2002) from the seawater entering the SOC. Many larger benthic foraminifera are hyaline species although the amount of Mg in their shells is often more than 10 times higher than that of planktonic and small benthic hyaline species, hence covering a large range in Mg/Ca values.

The calcification strategy of porcelaneous foraminifera is less well studied, which may be partly explained by their limited application in paleoceanography. Porcelaneous foraminifera use a different mode of calcification (Berthold, 1976; Hemleben et al., 1986; Debenay et al., 1998; De Nooijer et al., 2009) and produce shells without pores (hence, the term imperforate) consisting of tablets or needles (Debenay et al., 1998; Erez, 2003; Bentov and Erez, 2006). These calcitic needles (2-3µm) are precipitated intracellularly (Berthold, 1976), after which they are transported out of the foraminifer to form a new chamber (Angell, 1980). At the outer and inner layers of these chambers, the needles are arranged along the same orientation so that they form an optically homogenous surface, giving it a shiny (hence the term 'porcelaneous') appearance. In general the Mg/Ca values of the shells of porcelaneous foraminifera are high.

Remarkably, despite this large biological control, incorporation of minor and trace elements still reflects environmental conditions, in both hyaline and porcelaneous foraminiferal shells. For instance, the Mg/Ca of foraminiferal shells is primarily determined by seawater temperature (Allen and Sanders, 1994; Nürnberg et al., 1996) and seawater Mg/Ca (Chapter 3; Segev and Erez, 2006; Evans et al., 2015; Wit et al., in review). After correcting for the effect of the latter (if necessary) the use of foraminiferal Mg/Ca has been validated by its wide application as paleothermometer (Elderfield and Ganssen, 2000; Lear et al., 2000). Insight in vital effects (Erez, 2003) and inter-specific differences in trace element incorporation (Bentov and Erez, 2006; Toyofuku et al., 2011; Wit et al., 2012) is needed for making the Mg/Ca thermometer more robust. Systematic offsets between different species, interdependence of trace elements incorporated (Langer et al., in press.) and the different response of element incorporation on



element speciation (Chapter 6; Keul et al., 2013; Wit et al., 2013), potentially provides useful clues for
determining which processes play an important role in the biomineralization pathways.
Here we present the results from a controlled growth experiment for which we used several
(intermediate- and high-Mg) hyaline and miliolid species and an inter-species comparison of trace
elements. We assessed the impact of bio-calcification on element incorporation as a function of $p$CO$_2$
in order to contrast the impact of different calcification strategies. During foraminiferal calcification,
incorporation of certain elements or fractionation of certain isotopes is shown to depend on the carbonate
system, e.g. U/Ca$_{CALCITE}$ (Russell et al., 2004; Keul et al., 2013) and Zn/Ca$_{CALCITE}$ to [CO$_3^{2-}$] (Marchitto
et al., 2000; Chapter 6) and δ$^{11}$B to pH (Sanyal et al., 1996). Species-specific differences in partitioning
and fractionation most likely primarily reflect differences in calcification strategy. Offsets are largest
between hyaline and miliolid species, due to their fundamentally different calcification strategies (see
for a summary, Toyofuku et al., 2011). Differences in chemical composition and their dependency on
environmental variables can hence be used to identify key processes in miliolid and hyaline calcification.
We cultured eight benthic foraminiferal species (4 hyaline and 4 porcelaneous) under four different
$p$CO$_2$ conditions, analyzing incorporation of Mg, Sr, Na, Zn and Ba. Results are combined and compared
with literature data, to identify processes involved in calcification.

**2. Methods**
**2.1 Foraminiferal collection**
Large samples of macroalgae (*Dictyota* sp.) were collected in November 2015 at a depth of 2-3 meters
in Gallows Bay, St. Eustatius (N 17°28'31.6", W62°59'9.4"). Salinity was ~34 and temperature was
~29°C at the site of collection. The collected macroalgae were transported to the laboratory at the
Caribbean Netherlands Science Institute (CNSI), where they were placed in a 5 L aquarium with aerated
and unfiltered seawater. From this stock, small amounts of algae and debris were gently sieved over a
90 and 600 µm mesh to carefully dislodge foraminifera. Several species of foraminifera were picked
from the resulting 90-600 µm fraction. Living specimens of *Marginopora vertebralis* (Quoy & Gaimard,
1830), *Amphistegina gibbosa* (d'Orbigny, 1839), *Laevipeneroplis bradyi* (Cushman, 1930) and *Archaias*
*angulatus* and limited amounts (<20) of *Peneroplis pertusus* (Forskål, 1775)*, Asterigerina carinata*
(d'Orbigny, 1839), *Heterostegina antillarum* (d' Orbigny, 1839), and *Planorbulina acervalis* (Brady,
1884) characterized by yellow cytoplasm and pseudopodial activity, were selected for the culturing
experiments.

**2.2 Culture set-up**



We used an adapted version of the culture set-up described in Chapter 7. In short, four barrels each
containing 100 L of seawater (5µm filtered), were connected to a Li-Cor $CO_2/H_2O$ analyzer (LI-7000),
to regulate the $CO_2$ level in the barrels' head space. The set levels were maintained by addition of $CO_2$
and/ or $CO_2$-scrubbed air according to the monitored $p$CO$_2$. The set-points for $p$CO$_2$ were 350 (A), 450
(B), 760 (C) and 1400 (D) resulting in four batches of seawater differing only in their inorganic carbon
chemistry. Salinity (34.0±0.2) was monitored with a salinometer (VWR CO310). The fluorescent
compound calcein (Bis[N,N-bis(carboxymethyl)aminomethyl]-fluorescein) was added to the culture
media (5 mg/L seawater) to enable determination of newly formed chambers during the culture
experiment (Bernhard et al., 2004). Short-term exposure (<three weeks) to calcein has no detectable
impact on the physiology of benthic foraminifera (Kurtarkar et al., 2015), and the presence of calcein
has no effect on the incorporation of Mg and Sr in foraminiferal calcite (Dissard et al., 2009). Culture
media was stored air-free in portions of 250 ml in Nalgene bottles with teflon lined caps at 4°C until
further use.
Foraminifera were divided over the different treatments in duplicate and placed in 70 ml Falcon ® tissue
bottles with gas-tight caps in a thermostat set at 25°C (Fig. 1). The thermostat was monitored by a
temperature logger (Traceable Logger Trac, Maxi Thermal), monitoring the temperature every minute.
To create uniform light conditions, the thermostat was equipped with two LED shelfs, which resulted in
high light conditions 12 hr/12hr. Culture media was replaced every four days, to avoid build-up of
organic waste and to obtain stable seawater element concentrations and carbon chemistry. Foraminifera
were fed after every water change with 0.5 ml of concentrated freeze-dried *Dunaliella salina* cells, pre-
diluted with the corresponding treatment seawater. After 21 days, the experiment was terminated.
Foraminifera were rinsed three times with de-ionized water, dried at 40°C and stored in
micropaleontology slides until further analysis at the Royal Netherlands Institute for Sea Research
(NIOZ).

**2.3 Analytical methods**
**2.3.1 Seawater carbon parameters**
At the start and termination of the experiment, 125 ml samples of the seawater at each of the different
experimental conditions were collected to analyze dissolved inorganic carbon (DIC) and total alkalinity
(TA) on a Versatile INstrument for the Determination of Titration Alkalinity (VINDTA) at the CNSI.
Using the measured DIC and TA values and the software CO2SYS v2.1, adapted to Excel by Pierrot et
al. (2006) the other carbon parameters (including $[CO_3^{2-}]$ and $\Omega_{calcite}$) were calculated. For this we used
the equilibrium constants for K1 and K2 of Mehrbach et al. (1973), refitted by Dickson and Millero
(1987) (Table 1).




### 2.3.2 Seawater element concentrations

At the start and end of the experiment and during replacement of the culture media, subsamples were
collected in duplo using 50 ml LDPE Nalgene bottles and immediately frozen at $-80^0$C. After
transportation to the NIOZ, melted samples were acidified with 3 times Quartz distilled HCl to pH ~1.8
and the seawater composition of the samples was analyzed on an Element 2 sector field double focusing
mass spectrometer (SF-ICP-MS) run in medium resolution mode. IAPSO Standard Seawater was used
as a drift monitor. Analytical precision (relative standard deviation) was 3% for Ca, 4% for Mg, 1% Na,
1% for Sr and 5% Ba. We obtained average values of 5.25±0.06 mol/mol for Mg/Ca, 44.6±0.6 mol/mol
Na/Ca, 8.63±0.05 mol/mol for Sr/Ca, and 9.04±0.47 for μmol/mol Ba/Ca.
A subsample was analyzed using a commercially available pre-concentration system, SeaFAST S2.
With the SeaFAST system elements with low concentrations can be pre-concentrated to values above
detection limit of the SF-ICP-MS. Accordingly, we measured Cd, Pb U, B, Ti, Mn, Fe, Co, Ni, Cu, and
Zn. In short, 10ml of sample was mixed with an ammonium acetate buffer to pH 6.2 and loaded on a
column containing NOBIAS chelating agent. After rinsing the column with a diluted ammonium acetate
buffer the metals were eluted in 750 μl of quartz distilled 1.5 M $HNO_3$ before being quantified on the
SF-ICP-MS. Here we use the Zn data only, as this was analyzed in the foraminifera well. Analytical
precision (relative standard deviation) was 5% for Zn. We obtained average values 15.3±0.5 μmol/mol
for Zn/Ca for all treatments. Although these values are clearly above natural open ocean values, the
concentrations are very uniform between treatments and when comparing start and end of the
experiments. The contamination with Zn might hence have occurred already when filling the culture
setup with the waters from the bay adjacent to the culture facility. In any concentrations are well below
values considered toxic (Nardelli et al., 2016).

### 2.3.3 Cleaning methods

After termination of the experiment, foraminiferal shells were cleaned following an adapted version of
Barker et al. (2003). Per treatment duplicate, all foraminifera were transferred to 10 ml PE vials. To each
vial, 10 mL 1% $H_2O_2$ solution (buffered with 0.5M $NH_4OH$) was added to remove organic matter. The
vials were heated for 10 minutes in a water bath at 95 °C, and placed in an ultrasonic bath for 30 seconds
(degas mode, 80kHz, 50% power), after which the oxidizing reagent was removed. These steps (organic
removal procedure) were repeated five times. Foraminiferal samples were rinsed five times with
ultrapure water, after which the vials were stored overnight in a laminar flow cabinet at room
temperature to dry. Dried foraminifera were placed on double sided tape on LA-ICP-MS stubs. Pictures
were taken of individual foraminifera with a ZEISS Axioplan 2 fluorescence microscope equipped with



appropriate excitation and emission optics and a ZEISS Axiocam MRc 5 camera, to assess the number
of chambers added during the experiment based on the incorporation of calcein.

**2.3.5. LA-ICP-MS**
Element concentrations of individual fluorescent chambers were analyzed by Laser Ablation-ICP-MS
(Reichart et al., 2003; Van Dijk et al., in review). To determine foraminiferal element concentrations,
the laser system (NWR193UC, New Wave Research) at the Royal NIOZ was equipped with a 2-volume
cell 2 (New Wave Research), characterized by a wash-out time of 1.8 seconds (1% level) and hence
allowing detection of variability of obtained element to Ca ratios within chamber walls. Single chambers
were ablated in a helium environment using a circular laser spot with a diameter of 80 μm (*M.*
*vertebralis*) or 60 μm (other species). We ablated all calcein-stained chambers twice, except for the first
1-2 chambers that formed during the experiment to avoid contamination of calcite of chambers formed
prior to the experiments that may be overlapped by the first labelled chambers (Fig.2).
All foraminiferal samples were ablated with an energy density of $1\pm0.1$ J/cm$^{-2}$ and a repetition rate of 6
Hz. The resulting aerosol was transported on a helium flow through an in house build smoothing device,
being mixed with a nitrogen flow (2 L/min), before entering the quadrupole ICP-MS (iCAP-Q, Thermo
Scientific). Monitored masses included $^{7}$Li, $^{11}$B, $^{23}$Na, $^{24}$Mg, $^{25}$Mg, $^{27}$Al, $^{43}$Ca, $^{44}$Ca, $^{66}$Zn, $^{88}$Sr and $^{137}$Ba.
Contrary to $^{67}$Zn and $^{68}$Zn, $^{66}$Zn is free of interferences when measuring calcium carbonate and SRM
NIST glass standards (Jochum et al., 2012). Potential contamination or diagenesis of the outer or inner
layer of calcite was excluded by monitoring the Al signal. At the start of each series, we analyzed SRM
NIST612 and NIST610 glass standard in triplicate (using an energy density of $5\pm0.1$ J/cm-2), JCt-1
(coral carbonate) and two in-house standards, namely NFHS (NIOZ Foraminifera House Standard;
Mezger et al., in review) and the Iceland spar NCHS (NIOZ Calcite House Standard). We further
analyzed JCp-1 (Giant clam) and MACS-3 (Synthetic Calcium Carbonate) at the start of each series,
and to monitor drift after every ten samples. All element to calcium ratios were calculated with an
adapted version of the MATLAB based program SILLS (Guillong et al., 2008). SILLS was modified to
evaluate LA-ICP-MS measurements on foraminifera, allowing import of Thermo Qtegra software
sample list, laser data reduction and laser LOG files. Major adaptions include improved automated
integration and evaluation of (calibration and monitor) standards, quality control report of the monitor
standards and export in element to calcium ratios (mol/mol). Calibration was performed against the
MACS-3 carbonate standard, with $^{43}$Ca as an internal standard and we used the multiple measurements
of MACS-3 for a linear drift correction. Relative analytical precision (relative standard deviation (RSD)
of all MACS-3 analyses) is 3% for $^{23}$Na, 3% for $^{24}$Mg, 3% for $^{25}$Mg, 4% for $^{66}$Zn, 3% for $^{88}$Sr and 3%
for $^{137}$Ba. In total, 961 analyses were performed on 251 specimens covering eight species cultured in
four experimental conditions (see Table 2 for specifics).





We calculated the standard deviation (STD), RSD and standard error (STD/√n; SE) per treatment. The
partitioning coefficient (D) of an element (E) between seawater and foraminiferal calcite is expressed
as $D_E = (E/Ca_{CALCITE})/(E/Ca_{SW})$. Partition coefficients, element versus calcium ratio and growth
parameters were statistically compared with different experimental parameters (such as $pCO_2$ or $[CO_3^{2-}$
]) using a two-sided T-test with 95% confidence levels. This also allows for the calculation of 95%
confidence intervals over the average per treatment. Pairwise comparisons were made for per E/Ca per
species and culture conditions using ANOVA. Groups that showed significant difference were assigned
different letters. When comparing partition coefficients to other studies, $E/Ca_{SW}$ data was, in some
studies, not measured. In these cases, we used average seawater $E/Ca_{SW}$ to calculate $D_E$ (see also
supplementary Table 1).

**3. Results**
**3.1 Inter-species differences in element incorporation**
In Table 3 we present all the elemental data for the eight species investigated in this controlled $pCO_2$
culture experiment. $Mg/Ca_{CALCITE}$ of Mg in hyaline species varies between 25.9-141.3 mmol/mol
Mg/Ca. In contrast, $Mg/Ca_{CALCITE}$ of miliolid species ranges from 121.3-149.3 mmol/mol. This large
spread in foraminifera E/Ca of hyaline species is also observed for Sr (1.7-3.1 mmol/mol), Na (3.4-19.5
mmol/mol), Zn (9.0-97.0 μmol/mol) and Ba (2.7-20.1 μmol/mol), while miliolids only vary over a
narrow range (Sr = 2.0-2.2 mmol/mol; Na = 3.8-5.8 mmol/mol; Zn = 53.0-140.8 μmol/mol; Ba = 18.0-
29.0 μmol/mol). When comparing Mg incorporation to that of the other elements studied here (Ba, Zn,
Sr and Na) between species (treatment B; Table 3), we observe a positive relation between $D_{Sr}$
($p < 0.0025$), $D_{Na}$ ($p < 0.0005$), $D_{Ba}$ ($p < 0.05$) and $D_{Zn}$ ($p < 0.005$) for hyaline species (Table 4). In general
hyaline species are enriched similarly in all elements (Fig. 3). Compared to porcelaneous species, the
hyaline shell building species which incorporate the most Mg (>100 mmol/mol Mg/Ca) incorporate
more Na, and Sr, while incorporating less Zn and Ba. Element incorporation across miliolid species is
less variable then observed for hyaline species and in general partition coefficients for these species
seem closer to inorganic values (Fig. 3). Including data from literature (both culture and field
calibrations; see supplementary Table S1), preferable in which both Mg/Ca and at least one other
element (Na, Sr, Ba or Zn) is measured, shows that the relation based on the Caribbean species studied
here is also more general applicable when including more species ($D_{Sr}$ = $p < 0.005$; $D_{Na}$ = $p < 0.0005$; $D_{Ba}$
= $p < 0.005$; $D_{Zn}$ = $p < 0.01$), even though this compiled data (labeled 'All studies' in Table 4) covers a
wide range in environmental and experiment conditions.



### 3.2 Element/Ca as a function of ocean acidification

In both porcelaneous and hyaline species we find an increase of $Zn/Ca_{CALCITE}$ and $Ba/Ca_{CALCITE}$ with $pCO_2$, while foraminiferal Sr/Ca, Mg/Ca and Na/Ca remain similar across the experimental conditions (Fig. 4 and Table 5). Sensitivity of both foraminiferal Zn/Ca and Ba/Ca to changes in seawater $pCO_2$ differs between the studied porcelaneous and hyaline species. When $pCO_2$ changes from 350 to 1200 ppm, Zn/Ca of hyaline foraminifera increase by a factor of 3.7 (*A. carinata*) or 4.5 (*A. gibbosa*) while miliolid foraminiferal Zn/Ca increases only by 1.3 (*M. vertebralis*), 1.8 (*A. angulatus*) and 2.1 (*L. bradyi*). Also sensitivity of foraminiferal Ba/Ca to the same change in $pCO_2$ shows a similar pattern, with Ba/Ca of hyaline species increasing by a factor of 3.6 (*A. carinata*) or 3.7 (*A. gibbosa*), while miliolid species increase Ba/Ca only with a factor of 1.8 (*M. vertebralis*), 1.6 (*A. angulatus*) or 2.1 (*L. bradyi*).

### 4. Discussion

### 4.1 Trends in element incorporation

Both miliolid and hyaline foraminifera promote calcification by increasing their internal pH (De Nooijer et al., 2009). Still, they might use different mechanisms to take up the ions ($Ca^{2+}$ and $CO_3^{2-}$) necessary for chamber formation, which is reflected in the different trends observed here. Element incorporation in hyaline foraminifera is highly interdependent, i.e. species with increased Mg content also incorporate more Sr, Na, Ba and Zn (Fig. 3). This observation suggests that uptake of all these elements is controlled by the same process, which may be the transmembrane transport of calcium ions to the site of calcification. Such transport likely involves $Ca^{2+}$ channels (Nehrke et al., 2013), capable of transferring other ions, like e.g. Mg, Sr and Na (Hess and Tsien, 1984; Allen and Sanders, 1994; Sather, 2005). This may result in an interdependence between all these elements studied such as observed here for the hyaline species if the selectivity for $Ca^{2+}$ of these channels vary between species. In contrast, miliolid species, building porcelaneous shells show much less inter-species variation in element incorporation and ratios between incorporated elements is thus relatively similar between species (Fig. 3). This may be explained by calcification from an internal reservoir, such as intracellular vacuoles containing (modified) seawater (Hemleben et al., 1986; Erez, 2003). The fact that the Mg partitioning in this foraminiferal group is similar to the inorganic partition coefficient may indicate that the carbonate is directly precipitated from seawater, without major removal of $Mg^{2+}$ ions. The relative similarity in partition coefficients of other elements between miliolid species are generally in line with an inorganic-like calcite precipitation, with only minor alteration of the elemental composition of the calcifying fluid by ion channels.



**4.2 Effect of ocean acidification on Element/Ca**

For neither miliolid nor hyaline species, foraminiferal Mg/Ca, Na/Ca and Sr/Ca systematically change with $p$CO$_2$. The impact of pH (and/or [CO$_3^{2-}$]) on Mg/Ca$_{CALCITE}$ and Sr/Ca$_{CALCITE}$ in foraminifera has been the subject of discussion (e.g., Elderfield et al., 1996; Dissard et al., 2010). In low-Mg benthic species, both Mg/Ca$_{CALCITE}$ and Sr/Ca$_{CALCITE}$ do not seem to depend on inorganic carbon system parameters, e.g. pH or [CO$_3^{2-}$] (Allison et al., 2011; Dueñas-Bohórquez et al., 2011). However, for several planktonic species pH does influence Mg/Ca$_{CALCITE}$ and Sr/Ca$_{CALCITE}$ (Lea et al., 1999; Russell et al., 2004; Evans et al., 2016). The effect of pH on Sr/Ca$_{CALCITE}$ might be explained via increased growth rates due to pH-associated changes in [CO$_3^{2-}$] (Dissard et al., 2010). However, due to the limited experimental set-up, we are not able to disentangle the effects of the different carbon parameters in this study. Still, here we show that incorporation of Mg, Sr and Na of the selected larger benthic hyaline and miliolid foraminifera are not significantly impacted when cultured over a range of $p$CO$_2$ and thus [CO$_3^{2-}$] and pH values. Observed offsets in studies using acid titration (Lea et al., 1999; Russell et al., 2004; Dueñas-Bohórquez et al., 2011; Evans et al., 2016) to alter the carbonate system might be related to changes in alkalinity rather than $p$CO$_2$ or DIC. In the experimental setup here alkalinity was kept constant between the different treatments, but pH, DIC and carbonate ion concentration varied as a function of $p$CO$_2$.

In contrast, foraminiferal Zn/Ca and Ba/Ca are significantly impacted by $p$CO$_2$ for all species studied here (Table 5; Fig. 4). Although Hönisch et al. (2011) suggested that the impact of carbonate chemistry on Ba incorporation is negligible, their data does suggest a trend over the same interval in pH as studied here. In hyaline foraminifera, Zn/Ca and Ba/Ca increases more as a function of $p$CO$_2$ (factor of 3.7-4.5 and 3.6-3.7, respectively when $p$CO$_2$ increases from 350 to 1200 ppm) compared to the miliolid species (1.3-2.1 and 1.6-2.1 times, respectively). In the culture set-up used, increasing $p$CO$_2$ increases DIC, reduces pH and thereby decreases seawater [CO$_3^{2-}$]. Speciation of Zn, Ba and also other elements, like U (Keul et al., 2013), is primarily controlled by seawater [CO$_3^{2-}$]. Using the PHREEQC (Parkhurst and Appelo, 1999) and the standard llnl database, the speciation of all elements studied here (Mg, Na, Sr, Zn and Ba) for our different seawater treatments were modelled. We observed a decrease of free ions (Zn$^{2+}$ and Ba$^{2+}$) and an increase in Ba and Zn carbonate complexes (BaCO$_3^0$ and ZnCO$_3^0$), with increasing $p$CO$_2$ (Fig. 5), while the activity of Mg$^{2+}$, Na$^+$ and Sr$^{2+}$ remained unaffected. This suggests that element incorporation in foraminiferal calcite might be depending on the bioavailability of free ions, which in the case of Ba and Zn, changes with $p$CO$_2$.

**4.3 Speciation in the foraminiferal microenvironment**



During inorganic precipitation, carbonate complexes (e.g. $MgCO_3^0$) are easily incorporated into the
calcite crystal lattice. However, foraminifera build their test from ions available at the site of
calcification, which is well separated from the surrounding seawater (De Nooijer et al., 2009). During
calcification, $Ca^{2+}$ is proposed to be transported from seawater to the SOC via ion channels (Nehrke et
al., 2013). This so-called trans-membrane transport (TMT) through $Ca^{2+}$ channels has also been found
for other marine organisms, including coccolithophores (Gussone et al., 2006). These $Ca^{2+}$ channels may
not discriminate perfectly between Ca ions and elements like Sr and Ba (Allen and Sanders, 1994),
causing accidental transport of these elements into the SOC. How much of a certain element will enter
the SOC in this way, depends on 1) the selectiveness of the channels and the characteristics of the
transported ions, 2) the element to calcium ratio in the foraminiferal microenvironment and 3) the
concentration gradient between seawater and the SOC. For instance, ions such as $Mg^{2+}$ are heavily
fractionated against during TMT, which is reflected by the low $D_{Mg}$ found in most species. The large
range in Mg/Ca values in hyaline species suggests that TMT plays an important, but also variable, role
in calcification of these species. The availability of some free ions, like Ba and Zn, changes as a function
of $p$CO$_2$ due to the formation of carbonate complexes (Fig. 5). When Zn and Ba form stable complexes
with carbonate ions they are no longer available for (sporadic) transport through the $Ca^{2+}$ channels,
decreasing the availability at the site of calcification and subsequently, incorporation into the
foraminiferal calcite (Fig. 6).
In summary, the amount of Zn and Ba available at the site of calcification is proportional to the
concentration of the ratio between $Ca^{2+}$ and free $Zn^{2+}$ and $Ba^{2+}$ in the foraminiferal microenvironment.
In turn, the amount of free Zn and Ba ions in seawater is controlled by their respective concentration in
seawater concentration, as well as $[CO_3^{2-}]$. Foraminiferal Mg/Ca, Na/Ca and Sr/Ca is not detectably
affected by $[CO_3^{2-}]$, since these elements do not form carbonate complexes over the range of $[CO_3^{2-}]$
studied here.

### 4.4 Element incorporation in hyaline species

Between hyaline species, we observe simultaneous increases in all elements incorporated and this trend
is confirmed when including published data for other species compiled from previous studies (Fig. 4
and supplementary Table S1). Interestingly, the two hyaline species that are most enriched in all
elements studied (Mg, Na, Sr, Zn and Ba) are also the foraminiferal species with the largest average
adult test size (*H. antillarum* and *P. acervalis*) for which data is available (this study). The other hyaline
species, *A. carinata* and *A. gibbosa,* have considerably smaller maximum shell sizes and lower Mg/Ca,
Sr/Ca, etc. values.





Two processes involving these calcium channels could possibly explain the observed size trend in
hyaline species. First, larger foraminifera have a smaller surface area to volume ratio and, therefore,
proportionally less $Ca^{2+}$ channels, assuming the density of these channels per surface area remains
similar. This would imply that fewer channels need to transport more ions for a given volume of $CaCO_3$
precipitated, which may in turn, possibly reduce selectivity between $Ca^{2+}$ and other divalent cations.
Secondly, a larger foraminifer will need more overall $Ca^{2+}$ compared to smaller species for the
production of a single new chamber, since the volume of the chamber walls increases with the size of
the individual. This increased uptake of $Ca^{2+}$ from the microenvironment around the foraminifer, may
cause a lower concentration of $Ca^{2+}$ in the direct surroundings of the foraminifer compared to the other
ions, which may subsequently translate into an increased transport of ions other than $Ca^{2+}$ to the site of
calcification.
A consequence of these hypotheses is that juvenile or smaller adults should have lower partition
coefficients than fully grown adults. Although some studies have shown a size effect for several
elements (e.g. Elderfield et al., 2002), other studies show no major effect of size on element partitioning
(e.g. Friedrich et al., 2012; Evans and Müller, 2013). The moderate trend observed within species, in
comparison to the large differences observed here between species, may indicate that species control
channel density per surface area as a function of average shell size of the species. Alternatively, the
maximum size of a species may be accompanied by a difference in their calcification mechanism (e.g.
the relative contribution of TMT in element uptake) explaining inter-species differences in element
partitioning. From an evolutionary point of view the latter explanation seems more likely.

### 4.5 Mechanisms for element uptake in miliolid foraminifera

In contrast to hyaline species, the miliolid species build porcelaneous shells that show much less inter-
species variation in element composition (Fig. 3). While hyaline species calcify in a (semi-)enclosed
space, miliolids precipitate their calcite intracellularly in vesicles in which they promote calcification
by increasing pH (De Nooijer et al., 2009). This suggests that these species calcify directly from seawater
(Ter Kuile and Erez, 1987). The fact that the Mg partitioning is close to the inorganic partition coefficient
in this foraminiferal group (Fig. 3) reflects that the carbonate is directly precipitated from intracellular
seawater, without major alteration of the original $[Mg^{2+}]$. The relative similarity in partition coefficients
between different porcelaneous shell building species is in line with primarily inorganic precipitation,
with only minor alteration of the elemental composition of the calcifying fluid by ion channels.
However, the observed correlation between $p$CO$_2$ and Ba and Zn (Fig. 4) suggests that Ca channels still
play a (modest) role in supplying $Ca^{2+}$ to the miliolid SOC. The contribution of $Ca^{2+}$ through TMT is
likely smaller than in hyaline species, since they already obtain calcium by including seawater in their



calcification vesicle prior to calcite precipitation. The considerably smaller flux of transmembrane $Ca^{2+}$
compared to perforate species explains the observed lower sensitivity of e.g. foraminiferal Zn/Ca and
Ba/Ca to changes in seawater $[CO_3^{2-}]$ in miliolid species (Fig. 4). This approximately 2 times lower
sensitivity of porcelaneous foraminifera compared to hyaline species suggests that miliolid foraminifera
acquire half of the necessary $Ca^{2+}$ through Ca-channels, and the other half directly from vacuolized
seawater. Element incorporation in miliolid foraminifera will therefore be mainly governed by their
respective concentrations in seawater, and to a lesser extent by the selectivity for $Ca^{2+}$/ permeability for
other ions during TMT.

**5. Conclusions**
Trends in element incorporation in larger benthic foraminifera can be explained by a combination of
differences in calcification strategy and seawater chemistry. Carbonate ion concentration in seawater
determines bioavailability of some ions (e.g. $Zn^{2+}$ and $Ba^{2+}$), which are transported through Ca-channels
to the site of calcification. For hyaline foraminifera, we observed increased element incorporation for
larger species compared to smaller species, which can be explained by more intense activity of these
channels and the relative concentration in seawater during calcification. For miliolid foraminifera, only
half of the needed Ca is acquired through these $Ca^{2+}$ channels, while the other half is obtained by
including small vesicles of seawater, leading to element partitioning to be more in line with inorganic
calcite.

**Acknowledgments**
This research is funded by the NIOZ – Royal Netherlands Institute for Sea Research and the Darwin
Centre for Biogeosciences project "*Double Trouble*: Consequences of Ocean Acidification – Past,
Present and Future –Evolutionary changes in calcification mechanisms" and the program of the
Netherlands Earth System Science Center (NESSC). Great thanks to all participants of the 2015
foraminifera culture expedition at the CNSI, St. Eustatia and Esmee Geerken for support with the salinity
culture experiment at the NIOZ. Furthermore, we would like to thank Kirsten Kooijman for supplying
*Dunaliella salina* cultures, Patrick Laan and Karel Bakker for seawater analysis and Mariëtte Wolthers
for providing technical support with PHREEQC. Lastly, we thank Jan-Berend Stuut (NIOZ) for the
usage of the Hitachi TM3000 SEM (NWO grant 822.01.008 and ERC grant 311152).





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



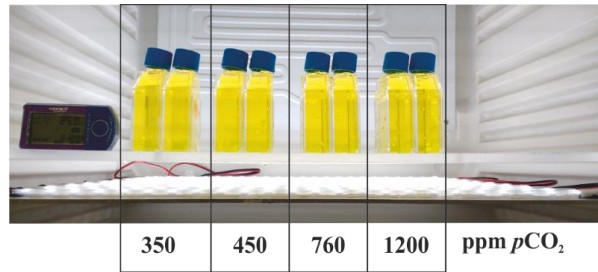


**Figure 1. Photograph of the culture set-up. Treatment with corresponding set-points are A=350,**
**B=450 ppm, C=760 ppm, D=1200 ppm.**



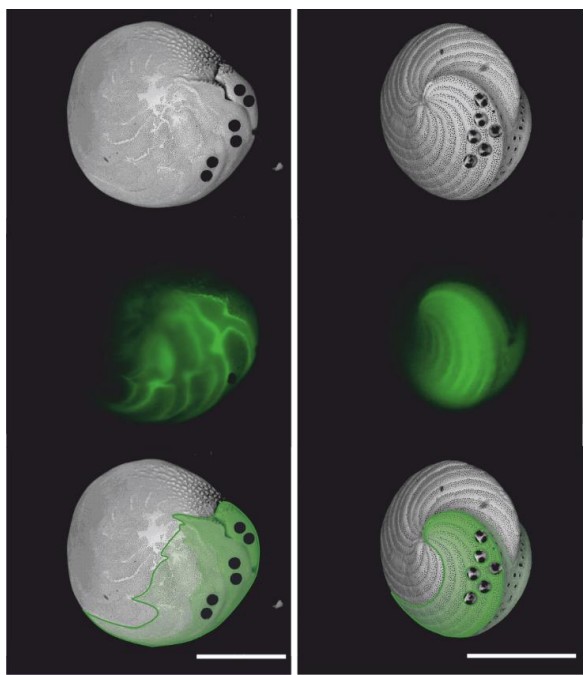


**Figure 2. SEM (top panels) and fluorescent microscope (middle panels) photographs of *A. gibbosa* (left) and *A. angulatus* (right) to assess newly formed chambers for laser ablation (lower panels). Scalebars = 500 μm.**



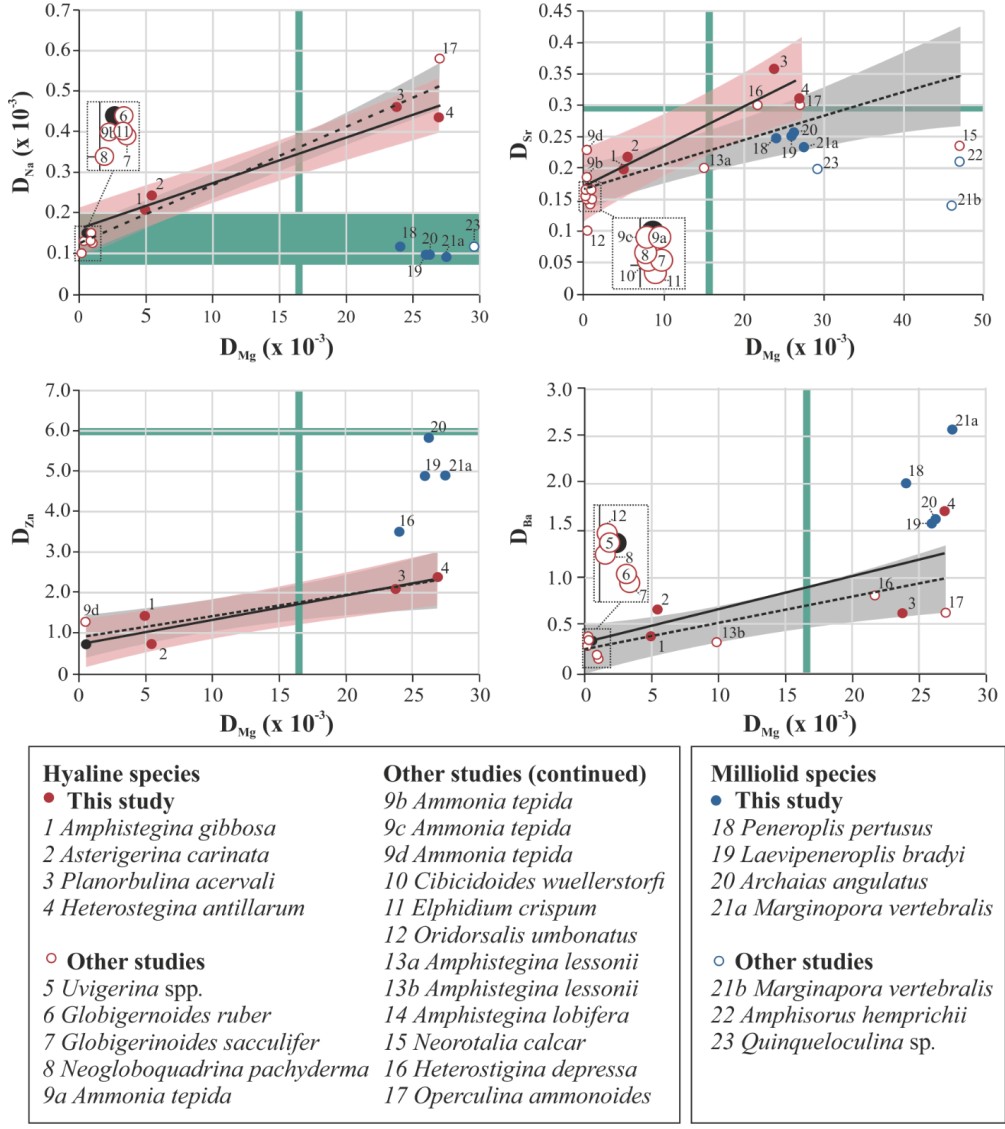


**Figure 3. Partition coefficient of Na, Sr, Zn and Ba versus $D_{Mg}$ of hyaline (red symbols) and miliolid (blue symbols) species in this study (closed symbols) and other studies (open symbols). Black lines represent trendlines (solid = this study; dashed = all studies). The 95% confidence intervals are indicated in pink (this study) and grey (all studies), which sometimes overlap. Black dots represent the NFHS, in-house carbonate standard, consisting of planktonic foraminifera. In green, inorganic partition coefficient from Mucci and Morse (1983), Ishikawa and Ichikuni (1984), Kitano et al. (1975) and Crocket and Winchester (1966). Numbers correspond to foraminiferal species analyzed (See supplementary Table S1)**





**Figure 4. Element to Ca ratios (±SE) of different species of foraminifera over a range of $p$CO$_2$ values. In some cases, the error bar is smaller than the symbol. Miliolid species in blue (triangles = *M. vertebralis*; squares = *A. angulatus*; circles = *L. bradyi*; squares = *P. pertusus*) and hyaline species in red (triangles = *H. antillarum*; squares = *P. acervalis*; circles = A. *carinata;* diamonds = *A. gibbosa*).**




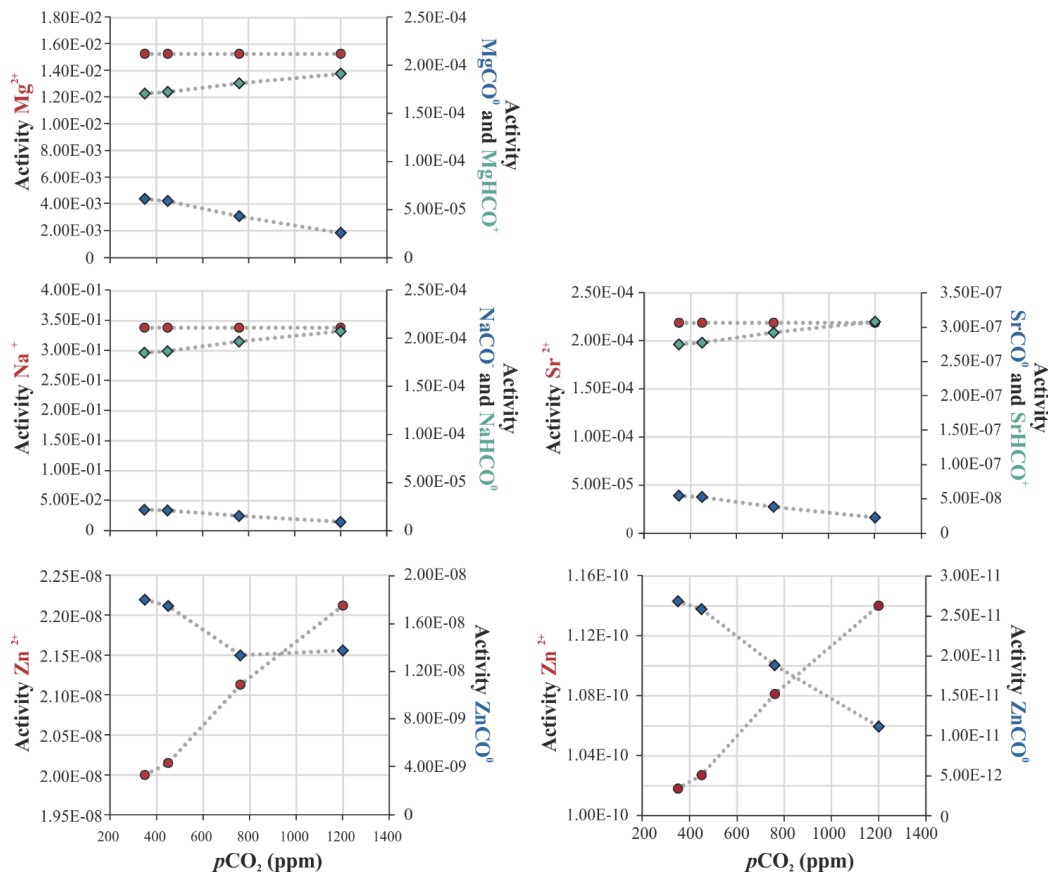

**Figure 5. Speciation of Mg, Na, Sr, Zn and Ba in the different seawater treatments modelled in**
**PHREEQC (Parkhurst and Appelo, 1999). Activities of free ions (red) and element (E)-carbonate**
**complexes (ECO$_3$ = blue diamonds and EHCO$_3$ = green diamonds).**



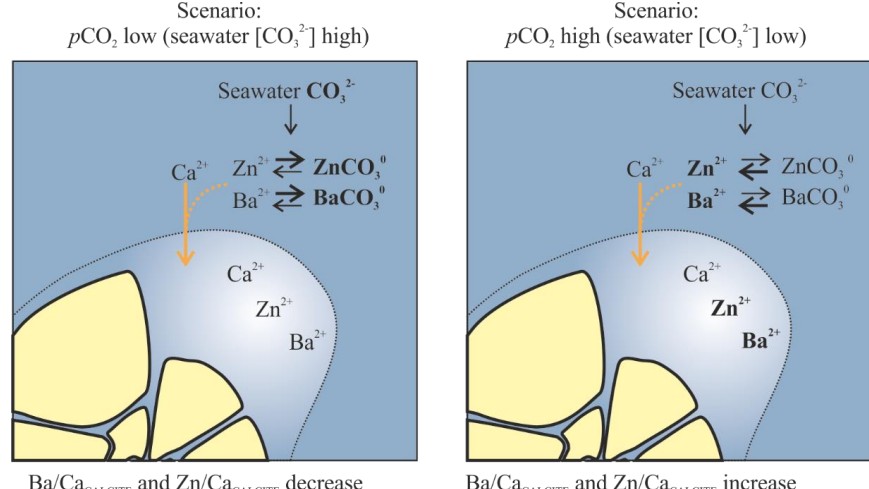

575   Ba/Ca$_{CALCITE}$ and Zn/Ca$_{CALCITE}$ decrease    Ba/Ca$_{CALCITE}$ and Zn/Ca$_{CALCITE}$ increase

576   **Figure 6. Schematic of incorporation of Zn and Ba during foraminiferal calcification under low**

577   **(left panel) and high (right panel) $pCO_2$ conditions. Amount of free ions (e.g. $Zn^{2+}$ and $Ba^{2+}$) is**

578   **influenced by speciation due to changing $[CO_3^{2-}]$. Orange arrow indicates transport of $Ca^{2+}$**

579   **through channel, with the associated accidental transport of $Zn^{2+}$ and $Ba^{2+}$.**






**Table 1. Carbon parameters (TA= Total alkalinity, n=2, DIC=Dissolved Inorganic Carbon, n=2)**
**with (relative) standard deviation of the culture water per treatment of the $p$CO$_2$ experiment.**
**CO2SYS was used to calculate seawater carbonate ion concentration, calcite saturation state and**
**pH from measured TA and DIC.**

| Treatment | Set-point | Measured | | Calculated CO2SYS | | |
|---|---|---|---|---|---|---|
| | $p$CO$_2$ ppm | TA µmol/kg | DIC µmol/kg | [CO$_3^{2-}$] µmol/kg | pH (total scale) | $\Omega_{CALCITE}$ |
| A | 350 | 2302.8±8.2 | 2007.5±10.7 | 211.2 | 8.04 | 5.1 |
| B | 450 | 2305.2±5.8 | 2021.3±12.5 | 204.1 | 8.02 | 4.9 |
| C | 760 | 2304.4±0.9 | 2100.8±13.4 | 153.3 | 7.86 | 3.7 |
| D | 1200 | 2300.3±0.7 | 2201.4±4.1 | 92.7 | 7.61 | 2.2 |




**Table 2. Total number of LA-ICP-MS measurements per species, per treatment (A-D).**

| Species | n LA(n specimens) | | | |
| --- | --- | --- | --- | --- |
| | **A: 350 ppm** | **B: 450ppm** | **C:760 ppm** | **D:1200 ppm** |
| *A. angulatus* | 62(19) | 72(21) | 76(21) | 51(14) |
| *M. vertebralis* | 48(14) | 49(15) | 57(18) | 33(11) |
| *A. gibbosa* | 106(28) | 126(32) | 75(18) | 59(15) |
| *L. bradyi* | 21(5) | 38(13) | 27(5) | 16(4) |
| *A. carinata* | 12(2) | 14(1) | 19(4) | 5(1) |
| *P. pertusus* | | 12 (2) | | 11 (2) |
| *H. antillarum* | | 12 (1) | | 14 (2) |
| *P. acervalis* | | 8 (2) | | |
| **Total** | 187(49) | 331(87) | 254(66) | 189(49) |




**Table 3. Overview of element to Ca ratios in foraminiferal calcite (Avg=average; SE=standard**
**error) and partition coefficients $D_E$, with $D_E$ of ambient conditions (treatment B) in bold. Letters**
**([a] to [d]) indicate (per species per E/Ca) groups that are statistical different (one-way ANOVA).**

| Species | $pCO_2$ | Mg/Ca mmol/mol Avg±SE | $D_{Mg}$ *10$^{-3}$ | Na/Ca mmol/mol Avg±SE | $D_{Na}$ *10$^{-3}$ | Sr/Ca mmol/mol Avg±SE | $D_{Sr}$ | Zn/Ca µmol/mol Avg±SE | $D_{Zn}$ | Ba/Ca µmol/mol Avg±SE | $D_{Ba}$ |
|---|---|---|---|---|---|---|---|---|---|---|---|
| *A. angulatus* | 350 | 139.4±0.6[a] | 26.6 | 5.2±0.1[a] | 0.12 | 2.2±0.02[a] | 0.25 | 80.0±5.1[a] | 5.3 | 13.2±0.5[a] | 1.5 |
| | 450 | 137.7±0.5[b] | **26.3** | 4.3±0.1[b] | **0.10** | 2.2±0.01[a] | **0.26** | 88.1±5.2[b] | **5.8** | 14.6±0.5[b] | **1.6** |
| | 760 | 137.4±0.7[b] | 26.2 | 4.9±0.1[c] | 0.11 | 2.2±0.01[a] | 0.26 | 122.6±7.0[c] | 8.1 | 17.0±0.6[b] | 1.9 |
| | 1200 | 138.6±1.1[a] | 26.4 | 5.4±0.2[a] | 0.12 | 2.2±0.02[a] | 0.26 | 140.8±9.9[d] | 9.3 | 20.9±0.2[c] | 2.3 |
| *M. vertebralis* | 350 | 147.7±0.6[a] | 28.2 | 4.8±0.1[a] | 0.11 | 2.0±0.01[a] | 0.24 | 70.0±10.1[a] | 4.6 | 17.0±0.5[a] | 1.9 |
| | 450 | 144.2±0.8[b] | **27.5** | 4.1±0.1[b] | **0.09** | 2.0±0.01[a] | **0.23** | 74.0±10.6[b] | **4.9** | 23.1±0.5[b] | **2.6** |
| | 760 | 143.0±0.6[a] | 27.3 | 3.8±0.1[a] | 0.09 | 2.0±0.01[a] | 0.23 | 87.7±15.5[c] | 5.8 | 27.9±0.6[c] | 3.1 |
| | 1200 | 148.3±0.5[b] | 28.3 | 4.5±0.2[c] | 0.10 | 2.0±0.01[a] | 0.23 | 115.6±15.3[d] | 7.6 | 30.1±0.2[d] | 3.3 |
| *L. bradyi* | 350 | 137.8±1.3[a] | 26.3 | 5.2±0.2[c] | 0.12 | 2.1±0.03[a] | 0.24 | 60.0±6.5[a] | 4.0 | 14.0±0.5[a] | 1.5 |
| | 450 | 136.2±0.7[a] | **26.0** | 4.3±0.1[b] | **0.10** | 2.2±0.01[b] | **0.25** | 73.8±6.0[b] | **4.9** | 14.2±0.5[a] | **1.6** |
| | 760 | 134.4±1.2[b] | 25.6 | 3.4±0.1[a] | 0.08 | 2.0±0.02[c] | 0.24 | 97.5±9.4[c] | 6.4 | 18.5±0.6[b] | 2.1 |
| | 1200 | 136.9±1.1[a] | 26.1 | 6.2±0.2[d] | 0.14 | 2.1±0.02[a] | 0.24 | 124.2±7.8[d] | 8.2 | 28.8±0.2[c] | 3.2 |
| *P. pertusus* | 350 | | | | | | | | | | |
| | 450 | 126.1±1.8[a] | **24.0** | 5.2±0.3[a] | **0.12** | 2.1±0.07[a] | **0.25** | 53.0±10.8[a] | **3.5** | 18.0±0.5[a] | **2.0** |
| | 760 | | | | | | | | | | |
| | 1200 | 121.3±1.0[a] | 23.1 | 5.8±0.2[a] | 0.13 | 2.2±0.02[a] | 0.26 | 75.5±11.9[b] | 5.0 | 29.8±0.2[b] | 3.3 |
| *H. antillarum* | 350 | | | | | | | | | | |
| | 450 | 141.3±0.3[a] | **26.9** | 19.4±0.5[a] | **0.44** | 2.7±0.02[a] | **0.31** | 36.0±14.7[a] | **2.4** | 10.7±0.5[a] | **1.2** |
| | 760 | | | | | | | | | | |
| | 1200 | 136.9±16[a] | 26.1 | 19.5±0.4[a] | 0.44 | 2.7±0.02[a] | 0.31 | 97.0±18.3[b] | 6.4 | 20.1±0.2[b] | 2.2 |
| *P. acervalis* | 350 | | | | | | | | | | |
| | 450 | 139.1±1.2 | **26.5** | 19.5±0.7 | **0.44** | 3.1±0.02 | **0.36** | 31.6±6.6 | **2.1** | 11.3±0.5 | **1.3** |
| | 760 | | | | | | | | | | |
| | 1200 | | | | | | | | | | |
| *A.* | 350 | 23.6±1.5[a] | 4.5 | 9.9±0.4[a] | 0.22 | 1.8±0.02[a] | 0.21 | 9.0±2.6[a] | 0.6 | 3.2±0.5[a] | 0.4 |





|  |  |  |  |  |  |  |  |  |  |  |  |
|---|---|---|---|---|---|---|---|---|---|---|---|
|  | 450 | 28.5±2.4$^b$ | **5.4** | 10.8±0.1$^a$ | **0.24** | 1.9±0.01$^a$ | **0.22** | 10.9±5.5$^a$ | **0.7** | 6.0±0.5$^b$ | **0.7** |
|  | 760 | 33.1±1.2$^b$ | 6.3 | 10.9±0.2$^a$ | 0.24 | 1.8±0.01$^a$ | 0.21 | 30.7±7.0$^b$ | 2.0 | 8.5±0.6$^c$ | 0.9 |
|  | 1200 | 33.5±3.1$^b$ | 6.4 | 10.6±0.5$^a$ | 0.24 | 1.8±0.03$^a$ | 0.21 | 46.4±2.1$^b$ | 3.1 | 11.4±0.2$^d$ | 1.3 |
| *A. gibbosa* | 350 | 27.8±0.5$^a$ | 5.3 | 9.0±0.1$^a$ | 0.20 | 1.7±0.01$^a$ | 0.20 | 19.0±1.8$^a$ | 1.3 | 2.7±0.5$^a$ | 0.3 |
|  | 450 | 25.9±0.6$^b$ | **4.9** | 9.2±0.1$^a$ | **0.21** | 1.7±0.02$^a$ | **0.20** | 21.5±2.5$^b$ | **1.4** | 3.4±0.5$^a$ | **0.4** |
|  | 760 | 28.2±0.7$^a$ | 5.4 | 9.7±0.1$^b$ | 0.22 | 1.7±0.02$^a$ | 0.20 | 52.8±6.1$^c$ | 3.5 | 7.1±0.6$^b$ | 0.8 |
|  | 1200 | 28.7±0.6$^a$ | 5.5 | 9.6±0.1$^b$ | 0.21 | 1.7±0.02$^a$ | 0.20 | 85.8±11.3$^d$ | 5.7 | 9.9±0.2$^c$ | 1.1 |






**Table 4. $R^2$ and p-values of linear trendline of $D_E$ versus $D_{Mg}$ of all hyaline species of this studies**
**and compiled literature studies (all studies).**

| $D_E$ versus $D_{Mg}$ | | $R^2$ | p-value |
|---|---|---|---|
| $D_{Na}$ | This study | 0.97 | <0.0005 |
| | All studies | 0.95 | <0.0005 |
| $D_{Sr}$ | This study | 0.90 | <0.0025 |
| | All studies | 0.53 | <0.005 |
| $D_{Zn}$ | This study | 0.88 | <0.005 |
| | All studies | 0.80 | <0.01 |
| $D_{Ba}$ | This study | 0.58 | <0.05 |
| | All studies | 0.56 | <0.005 |


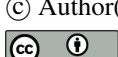


**Table 5. Regression and p-values of foraminiferal Zn/Ca and Ba/Ca versus $p$CO$_2$ values of**
**different species (Fig. 4).**

| Species | Zn/Ca | | Ba/Ca | |
|---|---|---|---|---|
| | $R^2$ | p-value | $R^2$ | p-value |
| *M. vertebralis* | 0.99 | <0.0005 | 0.81 | <0.025 |
| *A. angulatus* | 0.95 | <0.0025 | 0.99 | <0.0005 |
| *L. bradyi* | 0.98 | <0.0005 | 0.97 | <0.0025 |
| *A. carinata* | 0.98 | <0.001 | 0.94 | <0.005 |
| *A. gibbosa* | 0.99 | <0.0005 | 0.98 | <0.001 |
