# Peer review of "Contrasting trends in element incorporation in hyaline and miliolid foraminifera"

_Biogeosciences, 2016_

## Referee Comment (RC1) · K. Fujita (Referee) · 26 Oct 2016

A paper by van Dijk et al. reported different trends in elemental incorporation between rotaliid and miliolid foraminifers. In addition, this paper showed increasing trends of some (Zn, Ba) elemental incorporation with high pCO2 conditions, together with a chemical model that explain these trends. In particular, the latter results are attractive to many readers studying biocalcification and paleoceanography, with significant implications for proxy developments of paleo-pCO2. Although the manuscript includes novel data and ideas, is well written and structured, and seems suitable for publication in Biogeosciences, I suggest that authors consider the following points to improve the final version of the manuscript.

1) Authors should reconsider what is the main purpose (hypothesis), what is the main

results, and what are conclusions obtained from this study. In the Introduction, you may need more elegant story to explain why you conducted pCO2 controlled experiments as well as why you used symbiotic tropical large benthic foraminifera for this study, to understand trace element incorporation in foraminifera. I think the main important results in this study were increasing some trace element incorporation (Zn, Ba) with high pCO2 environments and a chemical model to explain this phenomenon.

2) Authors are mainly concerned about a correlation, but not dealt with the quantity (amount) of elemental incorporation in foraminiferal calcite. Even if both Zn and Ba ion availability increases with high pCO2 conditions, the amount incorporated in foraminiferal calcite differs between the two elements (Zn is four times more incorporated than Ba). We know that ionic radius is related to trace element incorporation in calcite from many inorganic studies. How does your chemical model explain the incorporation of elements quantitatively?

3) This paper is not discussed anywhere (only briefly mentioned in the Introduction) about effects of temperature on element incorporation in foraminifera, which are well established by many papers. In Figure 3, you compared results of your study with those of previous similar studies. However, I wonder these results are not simply comparable because of different species from different environments (water depths and latitudes, i.e. different temperature and salinity ranges) as shown in Supplementary Table S1. Temperature effects should be normalized to compare real DMg between different foraminiferal species. In addition, authors should discuss the relative sensitivity on E/Ca between temperature and carbonate chemistry when both parameters are variable.

4) This study assumes only Ca-channel model as a possible ion transport mechanism. However, other possible mechanisms of ion transport have been proposed in foraminifera. I suggest that authors compare advantages and disadvantages of several transport models and justify the Ca channel model as the most appropriate to explain results in this study.

5) I wonder if laser abrasion (LA) method is appropriate for biocalcification study. As you know, hyaline foraminiferal shells are composed of the primary layer and the secondary layers (coating) with the organic matrix. The LA method cannot discriminate differences in element incorporation between these different layers. In addition, the spatial heterogeneity of E/Ca among calcite crystals in a chamber wall has been reported by many studies. How do authors overcome these problems by using the LA method?

6) What kinds of other trace elements except for those examined in this study are sensitive to pCO2 based on the chemical model? That is useful information to find new proxies for paleo-pCO2. In addition, I wonder what cause differences between sensitive and insensitive elements on pCO2.

7) Authors are confused about the terminology of Foraminifera. Throughout the text, the authors used "hyaline" and "miliolid" as comparable terms. But the term "hyaline" indicates the quality of shell appearance and the term "miliolid" is a taxon name belonging to the Order Miliolida. I suggest authors use comparable terms of "hyaline vs. porcelaneous" as shell appearance, "perforate vs. imperforate" as shell perforation, and "rotaliid vs. miliolid" as the two main taxonomic group.

Specific comments are as follows:

L1: Title is vague and general, should be changed to include keywords and reflect the main results of this paper; for example, Calcification model of some trace element (Zn, Ba) incorporation in foraminifera under high pCO2 environments.

Abstract

L11-12: need an explanation why you conducted pCO2 controlled experiments for this study.

L13-19: How do you explain your results if other ion transport models are assumed?

L18: I think this is the main original results of this paper.

L22, due to differences in size: any other explanations possible (e.g., symbiont effects)?

Introduction

L38-40: This sentence is strictly speaking incorrect. Diverse miliolid foraminifers belonging to larger benthic foraminifera (LBF) are found particularly in the Atlantic and the Caribbean. In addition, LBF do not cover a large Mg/Ca range, but only intermediate and high Mg/Ca ranges. The authors have to explain advantages to use LBFs for their study in more detail.

L55: Is a paper in review OK to cite? If it is OK, it should be listed in the Reference.

L60: Not listed in the Reference.

L65: I do not understand why the authors conducted pCO2 control experiments for this study. Explain the rationale to conduct such experiments.

Methods

L86: 90-600 $\mu$m fraction is too small for larger benthic foraminifers (almost juveniles).

L86: As far as I know, Marginopora vertebralis (Quoy & Gaimard) is not distributed in the Caribbean and Atlantic (see Langer and Hottinger, 2000 Micropaleontology). Recheck if identification is correct.

L94: Where is "Chapter 7"? I also found other chapters in the text somewhere.

L97-98: Add pCO2 unit. Explain what (A) means.

L108-109: Add the precision of temperature control.

L110-111: Note the light intensity level. In addition, I wonder if LEDs and yellow culture bottles (Fig. 1) affect wave length and hence the growth of symbiotic foraminifers?

L113: Does food affect water quality and chemical composition?

L126: Delete "." before "K2"

L131: What is "duplo", meaning "duplicate"?

Section 2.3.2 to 2.3.5: These sections have many abbreviations that were not mentioned in the first appeared in the text. Probably this is because I am not familiar with geochemistry though, I suggest that authors explain some specific terms to readers who are not experts in geochemistry.

L131: LDPE?

L134: JASPO?

L138: SeaFAST S2?

L142: NOBIAS?

L150, (Nardeli et al., 2016): not listed in the Reference.

L154, Barker et al. (2003): not listed in the Reference.

L154: PE?

L155: Could the organic matrix in a shell be removed by this method? Does data not include any elemental incorporation in the organic matrix?

L167: What is the main difference between this paper and Van Dijk et al. (in review)?

L179-180: SRM NIST glass standards?

L182: NIST612, NIST610?

L187: MATLAB SILLS?

L189: LOG?

L192: MACS-3?

L202: T-test > t-test

Results

L211-218: Recheck numerical numbers of maximum/minimum values, that were not consistent with Table 3.

L217-218: Explain the rationale (hypothesis) why the authors compare Mg/Ca with other TE/Ca?

L218-219: not only significant values, but also $R^2$ values should be noted.

L223: Replace "then" to "than".

L227; DSr/DBa versus DMg are weakly correlated (Table 4), even though they are significant.

Discussion

L249-251: Compare advantages and disadvantages of several ion transport models and justify a Ca-channel model as the most appropriate to explain results in this study.

L260: I think miliolid foraminifera still need the major removal of Mg ions even if carbonate is directly precipitated from seawater.

L289: PHREEQC needs explanation

L290: llnl database?

Section 4.3: this section is mostly a review of previous studies. I suggest that authors explain an incorporation model shown in Fig. 6 in detail.

Section 4.4: Does size matter? authors mentioned that they measured only small size (L86). Calcarinids (Neorotalia in #15 in Fig. 3) are similar in size to Amphistegina, but have high Mg contents similar to a bigger Heterostegina. You may need another interpretation to explain the difference between two taxa. In addition, I think larger benthic foraminifers (in particular some taxa dwelling at a lower euphotic depth) have a strategy to attain a high surface area to volume ratio by flatting to get light for algal

symbionts. Please show the surface area to volume ratio between comparing taxa to justify your interpretations. Less Ca channels in the membrane of LBFs are also unlikely, because LBFs are bigger thus have much more membranes and channels than smaller foraminifers if channel density are the same. I think the second process is more feasible than the first process.

L347-350: I think this explanation is more plausible. Hyaline foraminifers are highly diverse and may have similar but slightly different calcification strategy acquired during evolution. I guess the relative contributions of primary and secondary layers and organic matrix may depend on hyaline foraminiferal taxa, which may cause interspecific variability of E/Ca compositions.

Section 4.5

L359: I think the major removal of Mg is necessary because your results show that DMg is much lower than 1.

L359-361: This sentence is a repetition of previous sentences.

L366-369: Is this correct? lower? I think higher or similar based on slope inclinations in Fig. 4. I do not understand how to estimate the relative contribution of seawater endocytosis and transmembrane transport. I guess some trans-membrane ion exchanges (Mg removal) occur between seawater vesicles and intrashell cytoplasm. High pCO2 seawater contains relatively large amounts of Zn and Ba ions, which are incorporated into foraminiferal cytoplasm via seawater vacuolization. Calcite needles are then precipitated from seawater vesicles with modifications by trans-membrane ion exchanges between seawater vesicles and intrashell cytoplasm.

Conclusions

These conclusions are not exactly what was found in this study and mostly speculative (hypothesis). I expect to see what this study revealed and what these results indicate.

Figure 6: Explain what bold types indicate.

Table 2: Better to write genus names in full in the table. A space between 450 and ppm.

Table 3: Better to indicate which species are either hyaline or porcelaneous.

[Figure]

---

## Referee Comment (RC2) · D. Evans (Referee) · 27 Oct 2016

The manuscript 'Contrasting trends in element incorporation in hyaline and miliolid foraminifera' by van Dijk *et al*. aims to investigate the response of the incorporation of certain trace elements to $pCO_2$ in eight species of miliolid and hyaline foraminifera. There is a large amount of high quality data here which will be of interest to the biomineralisation and proxy development community, and it is certainly suitable for publication in *Biogeosciences*. In particular, the miliolid data are interesting given that geochemical data from this group are sparse, and the discussion around the carbonate system impact on barium is an important finding. My comments are mostly minor, with the exception of those pertaining to the discussion on biomineralisation. As it stands the manuscript is written as if it is accepted that some ions for biomineralisation are sourced via trans-membrane transport (TMT). However, there is no current consensus

in the community even on the existence of this mechanism. Given that this dataset in isolation is not capable of resolving the issue, at the very least the discussion should explore the implications of these results for other biomineralisation models.

**Major comments**

1. Trans-membrane-transport is not required to explain these results, therefore a discussion of how they would be interpreted in terms of seawater vacuolisation should be included.

   - Although to my knowledge it remains to be tested, it is likely that the extent to which foraminifera raise the pH of seawater vacuoles is dependent on the ambient seawater pH (also by analogy to the ECF in corals). Therefore, seawater pH can be expected to influence (e.g.) Zn and Ba speciation in the seawater vacuole. Because a certain species is probably preferentially incorporated during crystal growth, there is no need to invoke poorly selective channels or pumps (lines 311-315).

   - Given that the Mg distribution coefficient for some species is greater than that of inorganic calcite, it is difficult to see how TMT helps explain the geochemistry of these foraminifera. In fact, it causes problems. If these species are still sourcing a portion of the Ca through channels or pumps (lines 309-311 and 362-364), then presumably the Mg/Ca ratio of the calcifying fluid is lower than that of seawater, yet in some cases they precipitate calcite with a Mg/Ca ratio ∼3 times that of inorganic calcite. There are three species shown in Figure 3 with a Mg/Ca ratio twice that of the highest Mg/Ca species of this study. A different mechanism is required here, and it is unclear how TMT could fit into this given it would require pumping Ca out of seawater to raise the Mg/Ca ratio before precipitation. In contrast to what is stated on

lines 357-359, the highest Mg species are equally (or even more) different from inorganic calcite as the low-Mg species.

2. Figure 3. The inorganic calcite distribution coefficients should only be displayed if they were characterised from calcite precipitated from seawater. For example, the sodium distribution coefficient is based on solutions with a chemistry very different from that of seawater, most notably the Mg concentration was much lower. It is coincidence that it is roughly the same as the miliolids and does not suggest that they precipitate shells with a similar $D_{Na}$ to that of inorganic calcite in seawater (lines 360-361). It is well known that Mg exerts a control on trace element distribution coefficients (see below), therefore it is not representative to compare these results to inorganic precipitation where these are carbonates precipitated from non-seawater solutions.

3. Section 4.1 and Figure 3. The reason that trace element distribution coefficients are strongly positively correlated with $D_{Mg}$ is because the incorporation of Mg into calcite modifies the incorporation of other elements through the associated lattice distortion. For example, this has been shown in inorganic calcite in the case of $D_{Sr}$ [Mucci & Morse, 1983] and $D_{Na}$ [Okumura & Kitano, 1986], and we confirmed that this is also the case in foraminifera through cultures in variable seawater Mg/Ca [Evans *et al.*, 2015]. The point is that this effect is not a consequence of ion transport, but has a basis in crystallography, especially given that hyaline foraminifera lie on the same $D_X$-Mg/Ca$_{calcite}$ line as inorganic precipitates [see Evans *et al.*, 2015 Fig. 7]. Furthermore, the trace element distribution coefficients shown in Figure 3 would be better expressed as a function of the calcite Mg/Ca ratio rather than $D_{Mg}$. It will not make much difference as most of these data are from foraminifera grown in seawater with a Mg/Ca ratio close to that of modern, but mechanistically it is the Mg concentration of calcite that is important.

**Minor comments**

1. Lines 60-61. Either there is a typo or this sentence needs rewording; I don't understand the intended meaning.

2. Line 69 and 94. Reference is made to thesis chapters which need updating.

3. Lines 268-270. It is true that some benthic species show little response of Mg/Ca and Sr/Ca to the carbonate system, but the $[CO_3^{2-}]$ effect on some deep benthic foraminifera Mg/Ca is well known. These are also low-Mg so this statement is not accurate.

4. Lines 277-279. There is no significant correlation between Mg/Ca and either DIC or alkalinity in these studies.

5. Lines 283-285. It is not really the case that there is a trend between Ba/Ca and the carbonate system in the planktonic cultures of Honisch *et al.* [2011]. Only the lowest pH cultures suggest any trend, but there are no replicates of these. How does the difference between these results and those presented here fit into the authors preferred biomineralisation model?

6. Line 289. Russell *et al.* [2004] should also be cited here.

7. Lines 334-335. I don't understand this statement. Do you mean that the selectivity of these channels depends on the amount of ions transported? Is there any evidence for this? It is more intuitive that selectivity is not changing.

8. Figure 3. There is a plotting mistake in the $D_{Na}$ panel. *P. acervali* and *H. antillarum* are shown with different sodium distribution coefficients but the data in Table 3 indiate that theyit is the same in both species.

9. Figure 5. I understand the logic for plotting this as a function of $p\text{CO}_2$, but given that pH is what we are able to reconstruct with boron isotopes I suggest adding a second set of $x$-axes to enable the two to be easily related. It would also be interesting to extend this plot to include the pH at the calcification site.

10. Figure 6. Half of this figure could be cut as both panels essentially show the same thing. Or, panel B could be replaced with a schematic showing how these results would fit into a biomineralisation model wherein the ions are sourced through seawater vacuolisation.

**Typos**

1. Line 12. This sentence is missing a word.

2. Line 81. Please add a unit to the number 34.

3. Lines 203-204. Consider rephrasing.

4. Lines 252-256. Consider rephrasing.

5. Figure 3. *P. acervalis* is spelt incorrectly.

---

## Author Comment (AC1) · 6 Dec 2016

Dear Dr Fujita,

Thanks for the thorough reading of our manuscript and your constructive comments

In general, the main concern of this referee was the focus of the manuscript, which should be more towards the potential of Zn/Ca and Ba/Ca as carbonate system proxies. By changing and restructuring the introduction and discussion session, we shifted the focus of the manuscript more towards the changes in Zn and Ba incorporation as a function of $p$CO$_2$. The conclusions and abstract are changed accordingly. Another one of the suggestions by both reviewers is to broaden the discussion by including other biomineralization and ion transport models to explain the species-specific trends in element incorporation.

The comments of the referee are posted below in italics. We address these issues point by point, in bold.

*General comments:*

*1) Authors should reconsider what is the main purpose (hypothesis), what is the main results, and what are conclusions obtained from this study. In the Introduction, you may need more elegant story to explain why you conducted pCO2 controlled experiments as well as why you used symbiotic tropical large benthic foraminifera for this study, to understand trace element incorporation in foraminifera. I think the main important results in this study were increasing some trace element incorporation (Zn, Ba) with high pCO2 environments and a chemical model to explain this phenomenon.*

**Based on this comment we now somewhat changed the focus of the introduction more towards carbonate system proxies (e.g. Zn/Ca) and the necessity to study this across different taxa since they are known to 1) have different calcification mechanism and 2) a different D for elements studied so far. We used tropical foraminifera since they have a higher variability in Mg incorporation and both porcelaneous and hyaline species are readily available in this region.**

**This also led to a reorganization of the discussion. First, we now discuss the Zn- and Ba-incorporation as a function of $p$CO$_2$ (now 4.2) and then discuss overall differences between hyaline and porcelaneous foraminifera (4.1 in the old manuscript). Then, we discuss the chemical speciation of ions as a function of $p$CO$_2$ (4.3). Finally, we evaluate existing biomineralization models based on our results.**

*2) Authors are mainly concerned about a correlation, but not dealt with the quantity (amount) of elemental incorporation in foraminiferal calcite. Even if both Zn and Ba ion availability increases with high pCO2 conditions, the amount incorporated in foraminiferal calcite differs between the two elements (Zn is four times more incorporated than Ba). We know that ionic radius is related to trace element incorporation in calcite from many inorganic studies. How does your chemical model explain the incorporation of elements quantitatively?*

**We now added these observations to the discussion. Following the TMT mixing model, selectivity of Ca$^{2+}$ channel differs for different elements, which probably also reflects in part ionic radius (see also, Nehrke et al., 2013).**

*3) This paper is not discussed anywhere (only briefly mentioned in the Introduction) about effects of temperature on element incorporation in foraminifera, which are well established by many papers. In Figure 3, you compared results of your study with those of previous similar studies. However, I wonder*

*these results are not simply comparable because of different species from different environments (water depths and latitudes, i.e. different temperature and salinity ranges) as shown in Supplementary Table S1. Temperature effects should be normalized to compare real DMg between different foraminiferal species. In addition, authors should discuss the relative sensitivity on E/Ca between temperature and carbonate chemistry when both parameters are variable.*

**We changed figure 3, to show the data of our study only. The previous figure 3 is now moved to the supplementary info, and we excluded data from foraminifera cultured/grown at low temperature, as they might partly have been grown under undersaturated conditions with repsect to calcite. Resulting species are all grown/cultured in a temperature range of 18-29 °C. We did not correct E/Ca for temperature, since the this effect on Na/Ca, Ba/Ca and Sr/Ca is not well constrained or unknown.**

*4) This study assumes only Ca-channel model as a possible ion transport mechanism. However, other possible mechanisms of ion transport have been proposed in foraminifera. I suggest that authors compare advantages and disadvantages of several transport models and justify the Ca channel model as the most appropriate to explain results in this study.*

**This issue was also raised by the second reviewer. The discussion is now put in a somewhat broader context, in which we also include other transport models, including seawater vacuolization (e.g. Ter Kuile and Erez, 1991; Erez, 2003, Elderfield et al., 1996; De Nooijer et al., 2014). This is added as a separated paragraphs in which we try to test/validate these models with our observations.**

*5) I wonder if laser abrasion (LA) method is appropriate for biocalcification study. As you know, hyaline foraminiferal shells are composed of the primary layer and the secondary layers (coating) with the organic matrix. The LA method cannot discriminate differences in element incorporation between these different layers. In addition, the spatial heterogeneity of E/Ca among calcite crystals in a chamber wall has been reported by many studies. How do authors overcome these problems by using the LA method?*

**Heterogeneity of elements in the chamber wall has only been observed in hyaline species, since they have (bi)laminar calcification. In this study we incubated adult foraminifera in culture media with calcein. LA-ICP-MS allows for targeting new chambers only, of which all layers of calcite were formed during the experiment. By using LA-ICP-MS we obtain an average signal of the chamber wall, averaging out any potential banding. Analysis of sufficient specimens/ chambers reduces uncertainty in average e.g. Mg/Ca values. Furthermore, we only ablated the final ~3 chambers, minimizing the potential effects of varying number of test carbonate layers. First paragraph of the discussion now lists this as one of the potential causes for element to calcium ratio (E/Ca) variability.**

*6) What kinds of other trace elements except for those examined in this study are sensitive to pCO2 based on the chemical model? That is useful information to find new proxies for paleo-pCO2. In addition, I wonder what cause differences between sensitive and insensitive elements on pCO2.*

**In this model we only looked at elements which we also measured by LA-ICP-MS. We modelleed the elemenst which based on their know geochemical behavour are the most likely to show differences as a function of changes in carbonate chemistry. All elements have been modelled in some sort of other study previously as well (e.g. Keul et al., 2013). When considering the impact of $[CO_3^{2-}]$ chemical speciation, the observed lack of sensitivity for Sr, Na and Mg might stem from their high concentration in seawater compared to e.g. Zn. Only a small amount of ions are hence complexed by $[CO_3^{2-}]$. Since there is a higher total amount of e.g. Mg ions in seawater, the amount of $Mg-CO_3$ complexation is relatively low. Due to their low concentrations and great affinity for**

**carbonate ions, elements like Cu, Co, Ni, Li, may be affected in the same way. We have added the possible behavior of these elements according to changes in speciation (4.3, chemical speciation).**

*7) Authors are confused about the terminology of Foraminifera. Throughout the text, the authors used "hyaline" and "miliolid" as comparable terms. But the term "hyaline" indicates the quality of shell appearance and the term "miliolid" is a taxon name belonging to the Order Miliolida. I suggest authors use comparable terms of "hyaline vs. porcelaneous" as shell appearance, "perforate vs. imperforate" as shell perforation, and "rotaliid vs. miliolid" as the two main taxonomic group.*

**To avoid any confusion, we changed the terminology in our paper to hyaline vs. porcelaneous and perforate vs. imperforate.**

L1: Title is vague and general, should be changed to include keywords and reflect the main results of this paper; for example, Calcification model of some trace element (Zn, Ba) incorporation in foraminifera under high pCO2 environments.

**We changed the title to: 'Trends in element incorporation in hyaline and porcelaneous foraminifera as a function of $p$CO$_2$' to better cover our main results.**

*Introduction:*

*L38-40: This sentence is strictly speaking incorrect. Diverse miliolid foraminifers belonging to larger benthic foraminifera (LBF) are found particularly in the Atlantic and the Caribbean. In addition, LBF do not cover a large Mg/Ca range, but only intermediate and high Mg/Ca ranges. The authors have to explain advantages to use LBFs for their study in more detail.*

**In our study we use intermediate to high Mg/Ca$_{CALCITE}$ (in our study ranging from 28.5 (*A. carinata*) to 141.3 (*H. antillarum*) mmol/mol), which is a large range in Mg/Ca$_{CALCITE}$.**

**New text: "A number of larger benthic foraminifera form hyaline shells, although the amount of Mg in their shells is often more than 10 times higher than that of planktonic and small benthic hyaline species, hence covering a larger range in Mg/Ca$_{CALCITE}$ values."**

*Methods:*

*L86: 90-600 µm fraction is too small for larger benthic foraminifers (almost juveniles).*

**We picked both directly from the macroalgae and from the 90-600 µm fraction, we now explain this in the methodology.**

*L86: As far as I know, Marginopora vertebralis (Quoy & Gaimard) is not distributed in the Caribbean and Atlantic (see Langer and Hottinger, 2000 Micropaleontology). Recheck if identification is correct.*

**We rechecked the identification of all of our species, and found that we misidentified *Sorites marginalis* as *Marginopora vertebralis*. This is now changed in the revised version of our manuscript.**

*L55: Is a paper in review OK to cite? If it is OK, it should be listed in the Reference.*

*L60: Not listed in the Reference.*

*L150, (Nardeli et al., 2016): not listed in the Reference.*

*L154, Barker et al. (2003): not listed in the Reference.*

*L94: Where is "Chapter 7"? I also found other chapters in the text somewhere.*

**5 points above: We removed all references to chapters and manuscripts in review**

*L97-98: Add pCO2 unit. Explain what (A) means.*

**We added the $p$CO$_2$ unit (ppm) and explain the treatment names A-D**

*L108-109: Add the precision of temperature control.*

**The average temperature over the whole experiment was 25±0.2°C, which is now added to the revised manuscript.**

*L110-111: Note the light intensity level. In addition, I wonder if LEDs and yellow culture bottles (Fig. 1) affect wave length and hence the growth of symbiotic foraminifers?*

**The culture bottles themselves are not yellow, it's the calcein added to the culture media. Almost all incubated foraminifera grew new chambers. We now added a table with the growth parameters.**

*L113: Does food affect water quality and chemical composition?*

**The described *dunaliella* feeding solutions do not change the chemical composition, since these dunaliella were rinsed, centrifuged, freeze-dried and subsequently diluted in the culture media. Water was replaced every four days to keep organic waste buildup at a minimum**

*L155: Could the organic matrix in a shell be removed by this method? Does data not include any elemental incorporation in the organic matrix?*

**In this cleaning step we remove the organic matrix from the foraminiferal shells. Although, in theory, it might be possible there are small amounts of matrix remaining in the carbonates, the amount of organic matter in foraminiferal shells is low, even before cleaning (e.g. 0.1-0.2 wt% of the total shell of *Heterostigina depressa*; wenier and Erez, 1984). However, for future work it might be interesting to analyze the organic matrix, to evaluate its potential contribution to the total E/Ca.**

*L167: What is the main difference between this paper and Van Dijk et al. (in review)?*

**We removed this reference, since this article is still in review. The differences with this study and Reichart et al. (2003) are summarized in this paragraph. For instance, we use different ICP-MS's, different cells, and used additional standards.**

Results:

*L217-218: Explain the rationale (hypothesis) why the authors compare Mg/Ca with other TE/Ca?*

**We included this figure in our manuscript because of the obseverd link between Mg and other TE published earlier (like e.g. Evans et al,., 2015).**

*L218-219: not only significant values, but also R2 values should be noted.*

**The R$^2$ values are presented in Table 3**

Discussion:

*L249-251: Compare advantages and disadvantages of several ion transport models and justify a Ca-channel model as the most appropriate to explain results in this study.*

*L260: I think miliolid foraminifera still need the major removal of Mg ions even if carbonate is directly precipitated from seawater.*

**We broaden the discussion to include also other transport models and how these might differ for porcelaneous and hyaline species, when comparing them to our observations.**

*L289: PHREEQC needs explanation*

*L290: llnl database?*

**PHREEQC and the llnl database are now explained in more detail in the new paragraph 4.3, which focusses on the modelling of chemical speciation.**

*Section 4.3: this section is mostly a review of previous studies. I suggest that authors explain an incorporation model shown in Fig. 6 in detail.*

**Due to a reorganization in our discussion, we now spent two paragraphs on different transport models.**

*Section 4.4: Does size matter? authors mentioned that they measured only small size (L86). Calcarinids (Neorotalia in #15 in Fig. 3) are similar in size to Amphistegina, but have high Mg contents similar to a bigger Heterostegina. You may need another interpretation to explain the difference between two taxa. In addition, I think larger benthic foraminifers (in particular some taxa dwelling at a lower euphotic depth) have a strategy to attain a high surface area to volume ratio by flatting to get light for algal C6 symbionts. Please show the surface area to volume ratio between comparing taxa to justify your interpretations. Less Ca channels in the membrane of LBFs are also unlikely, because LBFs are bigger thus have much more membranes and channels than smaller foraminifers if channel density are the same. I think the second process is more feasible than the first process.*

*L347-350: I think this explanation is more plausible. Hyaline foraminifers are highly diverse and may have similar but slightly different calcification strategy acquired during evolution. I guess the relative contributions of primary and secondary layers and organic matrix may depend on hyaline foraminiferal taxa, which may cause interspecific variability of E/Ca compositions.*

**We removed this part of our discussion, since we agree it is the less likely explanation of our observations for hyaline species. We end our discussion with a paragraph on the contribution of different mechanisms, which might explain our observations for both hyaline and porcelaneous species.**

*Section 4.5. L359: I think the major removal of Mg is necessary because your results show that DMg is much lower than 1.*

**With the seawater vacuolization model (Erez, 2003) it is indeed necessary to remove Mg ions from the calcification fluid. But the removal of Mg is not necessary when ions are transported by TMT,**

**since these channels mainly import Ca to the site of calcification. Only very few Mg ions would be transported (1 for every 10.000 Ca ions).**

*L366-369: Is this correct? lower? I think higher or similar based on slope inclinations in Fig. 4. I do not understand how to estimate the relative contribution of seawater endocytosis and transmembrane transport. I guess some trans-membrane ion exchanges (Mg removal) occur between seawater vesicles and intrashell cytoplasm. High pCO2 seawater contains relatively large amounts of Zn and Ba ions, which are incorporated into foraminiferal cytoplasm via seawater vacuolization. Calcite needles are then precipitated from seawater vesicles with modifications by trans-membrane ion exchanges between seawater vesicles and intrashell cytoplasm.*

**No, high $p$CO$_2$ contains the same amount of Zn and Ba ions, only speciation differs. Seawater vacuolized at different $p$CO$_2$ will have the same Zn and Ba concentration. The calcite needles which are precipitated from these vacuoles will have the same Zn/Ca and Ba/Ca, unrelated to ambient seawater carbonate chemistry. However, by combining TMT and seawater vacuolization, calcification fluid starts with ambient seawater, which is diluted with Ca$^{2+}$ by TMT. During this processes, the amount of ions other than Ca$^{2+}$ transported to the site of calcification depends on the chemical speciation (amount of free ions), the relative abundance compared to Ca$^{2+}$ and the selectivity of and thus discrimination by the Ca$^{2+}$ channels. This is now described in more detail in the discussion section.**

---

## Author Comment (AC2) · 6 Dec 2016

Dear Dr. Evans,

Thank you for the constructive comments we received on our manuscript. Below, all comments are listed in italics. We try to answer all specific comments raised by your review (in bold).

*Trans-membrane-transport is not required to explain these results, therefore a discussion of how they would be interpreted in terms of seawater vacuolisation should be included.*

**We agree with the reviewer that our results are not providing conclusive evidence for either one of the current biomineralization concepts, seawater endocytosis and TMT mixing. Still, the different observed correlations between the incorporation of trace elements in hyaline and porcelaneous species are best explained by primarily vacuolization in porcelaneous and a mixed signal in hyaline species. To accommodate the reviewers concern we now included a paragraph in which we evaluate the effect of the potential contribution of vacuolized seawater using our observations. The discussion now includes a comparison of the relative low (hyaline) and high (porcelaneous) contribution of vacuolized seawater on overall El/Ca.**

*• Although to my knowledge it remains to be tested, it is likely that the extent to which foraminifera raise the pH of seawater vacuoles is dependent on the ambient seawater pH (also by analogy to the ECF in corals). Therefore, seawater pH can be expected to influence (e.g.) Zn and Ba speciation in the seawater vacuole. Because a certain species is probably preferentially incorporated during crystal growth, there is no need to invoke poorly selective channels or pumps (lines 311-315).*

**How (if at all) the internal pH of foraminifera dependends on seawater pH outside the foraminifer, is currently not known. If it does change in concert, the internal pH at the site of calcification (which is ≥ 9; De Nooijer et al, 2009) would vary between >9 and >8.6 in our study. In theory such a change in internal pH from >9 to >8.6 changes $[CO_3^{2-}]$ and thus the speciation of e.g. Zn and Ba at the site of calcification. However, over this range the change in $[CO_3^{2-}]$ will be rather limited. Hence such an effect of differential speciation within the calcifying fluid does not suffice to explain the observed sensitivity in Zn and Ba to $pCO_2$ in our study. This is in line with recent evidence on Zn/Ca in foraminifera, which suggests Zn incorporation is not governed by changes in seawater pH, but by carbonate ion concentration which does not change very much anymore at these high pH's (van Dijk et al., 2017, figure 5d).**

*• Given that the Mg distribution coefficient for some species is greater than that of inorganic calcite, it is difficult to see how TMT helps explain the geochemistry of these foraminifera. In fact, it causes problems. If these species are still sourcing a portion of the Ca through channels or pumps (lines 309-311 and 362-364), then presumably the Mg/Ca ratio of the calcifying fluid is lower than that of seawater, yet in some cases they precipitate calcite with a Mg/Ca ratio ~3 times that of inorganic calcite. There are three species shown in Figure 3 with a Mg/Ca ratio twice that of the highest Mg/Ca species of this study. A different mechanism is required here, and it is unclear how TMT could fit into this given it would require pumping Ca out of seawater to raise the Mg/Ca ratio before precipitation. In contrast to what is stated on C2 lines 357-359, the highest Mg species are equally (or even more) different from inorganic calcite as the low-Mg species.*

**If the D of the foraminifera would be higher, this would indeed be true. However, we apologize for erroneously plotting the wrong inorganic D for Mg, which should have been ~150-200 (Mucci and Morse, 1983; Morse et al., 2007), indicating that all known foraminiferal partition coefficients are well below the inorganic D. We refrain from replotting this based on the next comment of this reviewer. Still, to accommodate this reviewers' concerns we extended our discussion, as mentioned above, to now also include other mechanisms potentially contributing to transport of ions to the SOC.**

2. Figure 3. The inorganic calcite distribution coefficients should only be displayed if they were characterised from calcite precipitated from seawater. For example, the sodium distribution coefficient is based on solutions with a chemistry very different from that of seawater, most notably the Mg concentration was much lower. It is coincidence that it is roughly the same as the miliolids and does not suggest that they precipitate shells with a similar DNa to that of inorganic calcite in seawater (lines 360-361). It is well known that Mg exerts a control on trace element distribution coefficients (see below), therefore it is not representative to compare these results to inorganic precipitation where these are carbonates precipitated from non-seawater solutions.

**We have removed the inorganic D's from figure 3, since the inorganically precipitated calcites are usually not derived from (natural) seawater. Most inorganic precipitation experiments do not utilize seawater as a source for carbonate precipitation, as it complicates the design of the experiment. This makes a direct comparison with foraminiferal calcite impossible.**

3. Section 4.1 and Figure 3. The reason that trace element distribution coefficients are strongly positively correlated with DMg is because the incorporation of Mg into calcite modifies the incorporation of other elements through the associated lattice distortion. For example, this has been shown in inorganic calcite in the case of DSr [Mucci & Morse, 1983] and DNa [Okumura & Kitano, 1986], and we confirmed that this is also the case in foraminifera through cultures in variable seawater Mg/Ca [Evans et al., 2015]. The point is that this effect is not a consequence of ion transport, but has a basis in crystallography, especially given that hyaline foraminifera lie on the same DX-Mg/Cacalcite line as inorganic precipitates [see Evans et al., 2015 Fig. 7]. Furthermore, the trace element distribution coefficients shown in Figure 3 would be better expressed as a function of the calcite Mg/Ca ratio rather than DMg. It will not make much difference as most of these data are from foraminifera grown in seawater with a Mg/Ca ratio close to that of modern, but mechanistically it is the Mg concentration of calcite that is important.

**We agree that Mg incorporation might distort the calcite crystal lattice, allowing for a higher incorporation of for instance Sr and Na. Some of us were actually actively involved in studies specifically targeting this (e.g. Mewes et al., 2015). In the new version of our manuscript we now emphasized this issue by add a paragraph (in 4.2), mentioning that this interdependency of element incorporation on Mg has been observed in both inorganic and culture experiments, and might partly stem from crystallography. However, although this mechanism might explain some of the observed species specific element incorporation in hyaline foraminifera, this does not explain the difference between hyaline and porcelaneous foraminifera. Porcelaneous foraminifera have in general high Mg/Ca, but we actually observe lower incorporation of Na and Sr compared to hyaline species with similar DMg (Fig 3, upper right and left panel: DMg versus DNa and DSr). When including porcelaneous species from other studies we also observe no increase in DSr over**

a larger range in DMg (Fig 3, upper left panel: DMg versus DSr). **This indicates that the mechanisms (or mechanisms) might be very different for hyaline and porcelaneous species, which is discussed in more detail at the end of our revised discussion.**

**We plotted partitioning coefficients rather than element to calcium ratios since a couple of the published studies changed the Mg/Ca of the culture media, and this makes it easier for readers to evaluate their own or other data.**

3. Lines 268-270. It is true that some benthic species show little response of Mg/Ca and Sr/Ca to the carbonate system, but the [CO2− 3 ] effect on some deep benthic foraminifera Mg/Ca is well known. These are also low-Mg so this statement is not accurate.

**For deep sea species benthic foraminifera, observed response to changes in carbonation ion concentration are mainly due to calcification in undersaturated seawater, as described by 'the carbonate ion saturation hypothesis' (Elderfield et al., 2006) and also observed for Zn by Marchitto et al., (2000, 2005) and Cd and Ba by McCorkle et al. (1995). We now added this references and a few sentences on the carbonate ion effect in undersaturated water to the revised discussion (4.1). However, there is some evidence that Sr incorporation is directly influenced by the carbonate system (also shown by Keul et al., in press). Still, sensitivity of Sr/Ca to e.g. $[CO_3^{2-}]$ is rather low (Keul et al., in press; van Dijk et al., 2017). In our experiment the change in $[CO_3^{2-}]$ between the highest and lowest $pCO_2$ treatment is rather limited (90-220 µmol/kg), resulting in no observed correlation of Sr/Ca with $pCO_2$**

4. Lines 277-279. There is no significant correlation between Mg/Ca and either DIC or alkalinity in these studies.

**We removed this statement from the discussion section and state there is a correlation between Mg/Ca and the carbonate system for planktonic foraminifera.**

5. Lines 283-285. It is not really the case that there is a trend between Ba/Ca and the carbonate system in the planktonic cultures of Honisch et al. [2011]. Only the lowest pH cultures suggest any trend, but there are no replicates of these. How does the difference between these results and those presented here fit into the authors preferred biomineralisation model?

**If incorporation of Ba depends on speciation due to differences in $[CO_3^{2-}]$, we expect plateauing of free $Ba^{2+}$ when we model speciation for lower pCO2 conditions (higher $[CO_3^{2-}]$). This is in line with the reviewers comments.**

7. Lines 334-335. I don't understand this statement. Do you mean that the selectivity of these channels depends on the amount of ions transported? Is there any evidence for this? It is more intuitive that selectivity is not changing.

**We have re-written this section of our discussion. We agree with the referee that the second explanation is more logical/intuitive, with the amount of $Ca^{2+}$ in the vicinity of the foraminifer decreasing, the relative element to calcium ratio of the seawater around the foraminifera increases. This would result in a higher transport of elements other than $Ca^{2+}$.**

*8. Figure 3. There is a plotting mistake in the DNa panel. P. acervali and H. antillarum are shown with different sodium distribution coefficients but the data in Table 3 indiate that theyit is the same in both species.*

**This is actually a mistake in the table. *P. acervali* has a DMg of 0.46 (aslo reflected in the higher Mg/CaCALCITE)**

*9. Figure 5. I understand the logic for plotting this as a function of pCO₂, but given that pH is what we are able to reconstruct with boron isotopes I suggest adding a second set of x-axes to enable the two to be easily related. It would also be interesting to extend this plot to include the pH at the calcification site.*

**We added a second x-axis to this figure, with the corresponding pH from our culture study. We are not able to plot pH at the calcification site, since it is unknown how internal pH responses to changes in ambient pH.**

10. Figure 6. Half of this figure could be cut as both panels essentially show the same thing. Or, panel B could be replaced with a schematic showing how these results would fit into a biomineralisation model wherein the ions are sourced through seawater vacuolisation.

**We adapted this figure to our revised discussion, and use it to show the contribution of both mechanisms to fit our observations.**